# The Relative Dose Intensity Changes during Cycles of Standard Regimens in Patients with Diffuse Large B-Cell Lymphoma

**DOI:** 10.3390/cancers15184458

**Published:** 2023-09-07

**Authors:** Shin Lee, Kei Fujita, Tetsuji Morishita, Eiju Negoro, Hikaru Tsukasaki, Kana Oiwa, Takeshi Hara, Hisashi Tsurumi, Takanori Ueda, Takahiro Yamauchi

**Affiliations:** 1Department of Hematology and Oncology, Faculty of Medical Sciences, University of Fukui, Fukui 910-1193, Japan; leesin@u-fukui.ac.jp (S.L.); kfujita@u-fukui.ac.jp (K.F.);; 2Department of Hematology and Oncology, Matsunami General Hospital, Gifu 501-6062, Japan; 3Department of Internal Medicine, Matsunami General Hospital, Gifu 501-6062, Japan; tmori@u-fukui.ac.jp; 4Department of Healthcare Economics and Quality Management, Graduate School of Medicine, Kyoto University, Kyoto 606-8501, Japan; 5Department of Cancer Care Promotion Center, Faculty of Medical Sciences, University of Fukui, Fukui 910-1193, Japan; 6Department of Hematology, Fukui Red Cross Hospital, Fukui 918-8501, Japan; 7Department of Internal Medicine, Osu Hospital, Nagoya 460-0017, Japan

**Keywords:** relative dose intensity, diffuse large B-cell lymphoma, cycle of chemotherapy, geriatric nutrition risk index, group-based trajectory modelling

## Abstract

**Simple Summary:**

The important role of relative dose intensity (RDI) of first-line chemotherapy in improving the prognosis of diffuse large B-cell lymphoma has been widely known, but no studies have focused on the trajectory of the average RDI (ARDI) during cycles. We explored the patterns of changes in ARDI during chemotherapy cycles and its association with overall survival. Maintaining a high ARDI in each cycle has a prognostic contribution to the outcome up to the sixth cycle, but no effect from the seventh cycle. The negative prognostic impact of significantly lower ARDI in the early part of the regimen could not be counteracted by increasing ARDI in the second half of the regimen. Malnutrition was associated with a significantly poor prognostic pattern of ARDI changes. A better understanding of ARDI trajectories may help with the early identification of deteriorating patients and has potential implications for the personalized prevention of reduced ARDI.

**Abstract:**

No studies have focused on the trajectory of the average relative dose intensity (ARDI) during cycles of first-line chemotherapy for patients with diffuse large B-cell lymphoma. To evaluate the impact of attenuating ARDI during cycles on overall survival, we conducted a multi-centre, longitudinal, observational retrospective study. A total of 307 analysable patients were enrolled. Multivariate Cox hazards modelling with restricted cubic spline models revealed prognostic benefits of higher ARDI up to, but not after, cycle 6. According to group-based trajectory modelling, patients were classified into five groups depending on the pattern of ARDI changes. Among these, two groups in which ARDI had fallen significantly to less than 50% by cycles 4–6 displayed significantly poorer prognosis, despite increased ARDI in the second half of the treatment period (log-rank *p* = 0.02). The Geriatric Nutritional Risk Index offered significant prediction of unfavourable ARDI changes (odds ratio 2.540, 95% confidence interval 1.020–6.310; *p* = 0.044). Up to cycle 6, maintenance of ARDI in all cycles (but particularly in the early cycles) is important for prognosis. Malnutrition is a significant factor that lets patients trace patterns of ARDI changes during cycles of chemotherapy associated with untoward prognosis.

## 1. Introduction

Diffuse large B-cell lymphoma (DLBCL) is the most common subtype of lymphoma [1,2]. Because of its high chemosensitivity, maintaining a high relative dose intensity (RDI) during first-line chemotherapy is crucial for improving the prognosis of DLBCL [3,4,5,6,7,8,9]. Reports on DLBCL to date have only addressed average RDI (ARDI) using time-averaged and/or cumulative ARDI [4,8]. Much less is known about the prognostic impact of longitudinal dynamic changes in ARDI over the entire chemotherapy period.

While the importance of ARDI for overall survival (OS) has been widely recognised, little is known about the number of cycles for which this importance is maintained. Further, there is a paucity of data stratifying the fluctuations in ARDI as chemotherapy cycles progress and focusing on the association with OS. We hypothesised that the survival benefits of maintaining a high ARDI in the early chemotherapy period would outweigh those in the late chemotherapy period, that multiple trajectory patterns would exist within ARDI fluctuations in the DLBCL population, and that certain trajectory groups would have a higher likelihood of death, with distinct patterns in clinical characteristics.

The aims of this study were thus: (1) to evaluate the impact of attenuating ARDI with the progression of cycles of standard first-line chemotherapy on OS for newly diagnosed DLBCL; (2) to identify subgroups of individuals with similar trajectories in ARDI during cycles of chemotherapy using group-based trajectory modelling (GBTM); and (3) to characterise clinical factors for those patient populations within each ARDI trajectory and determine associations between different trajectories and risk of death.

## 2. Materials and Methods

### 2.1. Study Population and Clinical Information

This retrospective, multicentre, observational, and longitudinal analysis was conducted at two tertiary institutions in Japan: University of Fukui Hospital, and the Japanese Red Cross Fukui Hospital. We reviewed the medical records and oncology pharmacy records of consecutive patients diagnosed with de novo DLBCL during the period from 2006 to 2021. Lymphoma was diagnosed according to the World Health Organization classification [2,10]. We defined the inclusion criteria as follows: patients with newly diagnosed and histologically proven de novo DLBCL; patients aged 18 years or older at the time of diagnosis; patients diagnosed with advanced-stage DLBCL or limited-stage DLBCL with any bulky mass; and patients receiving standard immunochemotherapy regimens as defined in the present study as first-line therapy. A previous phase III study showed that R-THP-COP (comprising rituximab [R], tetrahydropyranyl adriamycin [THP], cyclophosphamide [CPA], vincristine [VCR], and prednisolone [PSL]) is not inferior to R-CHOP (comprising R, adriamycin [ADR], CPA, VCR, and PSL) with regard to clinical response, and has an acceptable safety profile [11]. We defined the CHOP regimen and THP-COP regimen with or without R as standard immunochemotherapy regimens in the present study. We defined the exclusion criteria as follows: patients with central nervous system involvement; patients diagnosed with post-transplant lymphoproliferative disorder; patients diagnosed with methotrexate-related DLBCL; patients receiving treatment other than CHOP or THP-COP regimens; patients receiving fewer than six cycles of the first-line immunochemotherapy regimen; patients diagnosed with composite lymphoma consisting of DLBCL plus indolent lymphoma (transformed DLBCL); patients with human immunodeficiency virus infection; or cases with missing data.

The baseline demographics of patients were collected by retrospective medical chart review. Baseline characteristics including Eastern Cooperative Oncology Group performance status (PS), number of extranodal sites, International Prognostic Index (IPI), elevated lactate dehydrogenase (LDH) level, Ann Arbor stage, serum albumin level, B symptoms, bulky mass (maximum diameter > 7.5 cm), and soluble interleukin-2 receptor (sIL-2R) level were extracted. The Charlson Comorbidity Index (CCI) is a widely known comorbidity index [12]. A recent study reported the CCI as a host-dependent factor that correlates significantly with survival in patients with DLBCL [12,13]. We therefore collected the CCI at diagnosis to assess patient comorbidities. The Geriatric Nutritional Risk Index (GNRI) is a simple and validated scale for assessing malnutrition, consisting of serum albumin level and body mass index [14]. A previous study showed the prognostic utility of GNRI specifically for patients with DLBCL [15]. We have also reported that the GNRI not only offers a predictor of survival, but also shows markedly improved prognostic accuracy when incorporated with the CCI [14,16]. We therefore extracted the GNRI at diagnosis to evaluate the nutrition status of the patient.

### 2.2. Immunochemotherapy Regimens and Calculation of RDI

The CHOP protocol included CPA (750 mg/m^2^ intravenously on day 1), ADR (50 mg/m^2^ intravenously on day 1), VCR (1.4 mg/m^2^ [maximum, 2 mg/body] intravenously on day 1), and PSL (100 mg/body orally or intravenously on days 1–5). These agents were administered in a 21-day cycle. Except for THP taking the place of ADR, the THP-COP regimen was identical to the CHOP regimen in terms of dosage. Dose modifications and the timing of the start of subsequent cycles were decided at the discretion of the attending physician, depending on adverse events and the general condition of the patient.

Dose intensity (DI) is an index of the scheduled dose per specific period. DI was determined as the planned dose per course (mg/m^2^) divided by the planned period per course (weeks). To express the RDI as a percentage, the DI was divided by the corresponding target DI before being multiplied by 100. The average delivered RDI for each chemotherapeutic agent (ADR or THP, CPA, and VCR) for each cycle was defined as the ARDI [4]. Total ARDI (tARDI) was the average amount of ARDI supplied throughout all treatment cycles. A tARDI of 100% in the present study was defined as 6 cycles of R ± CHOP or THP-COP without any reduction in chemotherapeutic drugs or delay in treatment. As a result, the tARDI can exceed 100% in cases receiving more than 7 cycles of R ± CHOP or THP-COP without any reduction in chemotherapeutic drugs or delay in treatment interval.

### 2.3. Outcome Measures

We defined OS as the primary outcome of the present study. In addition, we evaluated prognostic factors thought to affect the OS of patients with DLBCL. OS was estimated from the date of diagnosis to the last follow-up appointment or date of death from any cause. Progression-free survival (PFS) was defined as the time from diagnosis until the first occurrence of disease relapse or progression or death from any cause. Patients who were still alive and showed no evidence of disease relapse or progression were censored as of the last follow-up. Retrospective chart reviews were used to correct all event dates, which were censored as of 27 December 2021.

Treatment response was assessed using whole-body computed tomography and/or 2-[18F]-fluoro-2-deoxy-D-glucose-positron emission tomography in patients who completed the treatment. Response was evaluated based on the revised response criteria for malignant lymphoma, with complete response (CR), partial response (PR), stable disease (SD), and progressive disease (PD) defined according to the 2007 revision of the criteria described by Cheson et al. [17]. In addition, unconfirmed CR was defined following the 1999 criteria of Cheson et al. [18].

### 2.4. Sensitivity Analysis

As post hoc sensitivity analyses, we also assessed the impact of attenuating ARDI with the progression of cycles on OS, limiting the data to the population of patients treated with R. We then also performed a sensitivity analysis to assess the impact of attenuating ARDI with the progression of cycles on PFS.

### 2.5. Statistical Analysis

We divided all patients into two groups, survivors and deceased, and compared background characteristics at diagnosis. Continuous variables are presented as median values and ranges and were compared between groups using the Mann–Whitney U test. Categorical variables are presented as numbers and percentages and were compared between groups using the chi-square test or Fisher’s exact test. Multivariate Cox hazards modelling with non-linear regression models with 3 knots restricted cubic splines (RCS) was used to evaluate the non-linear relationship in ARDI for each cycle of chemotherapy in both the survival and deceased groups [19]. Multivariate Cox-RCS models were also used to assess the presence of a non-linear relationship between ARDI and all-cause mortality, which varies with the cycle of chemotherapy. Multivariate non-linear regression with the Huber–White robust sandwich estimator of a variance–covariance matrix was used to correct for heterogeneous variance and for correlated responses from repeatedly measured values.

GBTM was used to identify patterns of ARDI change during the cycle of chemotherapy. The optimal number of trajectories was selected to fit the data, as evaluated by the Bayesian information criterion and the percentage of individuals attributed to each trajectory group. Survival curves for each cluster were estimated by the Kaplan–Meier method, and log-rank testing was used for comparisons between clusters. Multivariate logistic regression analysis was performed to identify factors associated with the presence of a pattern of favourable prognostic ARDI change as identified by GBTM. Multivariable adjustments were performed for age, sex, Eastern Cooperative Oncology Group-performance status (ECOG-PS), serum LDH, Ann Arbor stage, number of extranodal sites, CCI, and GNRI (a priori indicated traditional prognostic factors of DLBCL), with a multivariable Cox proportional hazards model for OS and a multivariate logistic regression model [13,16,20]. Among these covariates, age, LDH, ECOG-PS, Ann Arbor stage, and number of extranodal sites are prognostic factors used for the IPI. Thus, the selection of covariates includes all factors used for IPI risk classification. Model building and variable selections were based on the published DLBCL risk algorithm (including the IPI component) and substantive knowledge regarding DLBCL prognosis to guide variable selection. Covariates were determined a priori after our review of the literature and group meetings with our research staff in order to avoid the consequences of over-fitting. In the present study, all *p* values were two-tailed, with values of *p* < 0.05 considered significant. Data analysis was performed using R version 4.2.1 or EZR version 1.55, which is a graphical user interface for R [21,22]. GBTM was realised using the “latrend” package.

## 3. Results

### 3.1. Patient Characteristics

A total of 220 individuals who met any of the exclusion criteria were eliminated after 527 patients had been initially identified. There were thus 307 analysable patients (Appendix A). Among the total of 307 patients, 305 received standard chemotherapy regimens with R. Table 1 shows the comparison of patient backgrounds between the deceased and survival groups. Median age at baseline was 71 years (range, 16–96 years) and 92 deceased patients (29.9%) were confirmed. Patients in the survival group were significantly younger, with better PS, lower LDH, lower frequency of advanced-stage DLBCL, lower frequency of multiple extranodal sites, lower frequency of lesions, lower frequency of B symptoms, lower frequency of bulky masses, lower IPI, lower sIL-2R level, higher total ARDI, lower CCI, and higher GNRI.

### 3.2. Multivariate Cox-RCS Models for ARDI Changes

The median duration of follow-up was 40.7 months (range, 4.4–167.8 months). During this time, 92 patients died (29.9%), including 53 deaths (57.6%) due to lymphoma. The remaining 25 deceased patients (27.1% of total deceased patients) had another cause, or the cause of death was not recorded (*n* = 14; 15.2% of total deceased patients). Other causes of death were as follows: nine deaths due to infection, six deaths due to other cancers, four deaths due to haemorrhage (brain and lung), three deaths due to heart failure, two deaths due to exacerbation of chronic obstructive pulmonary disease, and one death by suicide. The breakdown of the results in terms of treatment response at the end of treatment was as follows: 273 patients obtained CR, and 30 patients obtained PR. The overall response rate (CR + PR) was thus 98.7% (303/307).

To evaluate the pattern of ARDI changes during the cycle of chemotherapy, we created multivariate Cox-RCS models of two groups: the survival and deceased groups (Figure 1). The survival group maintained a high ARDI of approximately 85% or more after the second cycle. Conversely, ARDI in the deceased group declined from the second cycle onwards, indicating that treatment intensity was not maintained at a constant level throughout the treatment period.

Multivariate Cox-RCS models for the relationship between mortality risk and ARDI of each cycle of treatment were created to assess whether the prognostic impact of ARDI varies from cycle to cycle (Appendix A). Age, sex, PS, elevated LDH level, number of extra-nodal sites, CCI, GNRI, and tARDI through the initial cycle to the cycle represented in each figure were used as covariables. Cycles 1–2, 3–4, and 5–6 showed a trend toward lower risk of death as ARDI increased. Cycles 7–8 displayed the opposite slope to the other cycles, with an increasing ARDI appearing to increase the risk of death.

### 3.3. Sensitivity Analysis for Survival Outcomes

The post hoc sensitivity analyses that limited the data to a population of 305 DLBCL patients treated with R also confirmed the results of multivariate Cox-RCS models for ARDI changes. We also conducted a sensitivity analysis that assessed the impact of attenuating ARDI with the progression of cycles on PFS. Multivariate Cox-RCS models for the relationship between PFS and ARDI changes identified similar results to those for OS (Appendix A).

### 3.4. Group-Based Trajectory Modelling for a Pattern of ARDI Changes

According to GBTM, five distinct trajectory groups were identified among the enrolled patients, depending on the pattern of ARDI changes during the cycle of chemotherapy (Figure 2). Group A (*n* = 184) had the highest number of patients, started the initial treatment cycle with a high ARDI close to full dose, and maintained that high ARDI throughout the entire treatment period. Group B (*n* = 39) started the initial treatment cycle with a high ARDI close to the full dose but lowered the ARDI after the second cycle. Group C (*n* = 48) started the initial treatment cycle with an ARDI of approximately 70 to 80% and maintained that ARDI throughout the entire treatment period. Group D (*n* = 22) included the lowest number of patients, started the initial treatment cycle with an ARDI of approximately 80%, and exhibited a steep decline in ARDI to less than 50% between cycles 2 and 6, but an increased ARDI after cycle 6. In Group D, a total of 21 patients (21/22, 95.5%) achieved CR or PR after completing 6 cycles of standard chemotherapy. Among these, 13 patients increased treatment intensity in the latter part of the overall chemotherapy period. All of these patients experienced severe adverse events during the initial treatment phase, but adverse events became manageable in the latter part of treatment, allowing for the escalation of ARDI. Group E (*n* = 14) started the initial treatment cycle with a low ARDI (less than 50%) that gradually increased from the fourth cycle onwards.

### 3.5. Survival Curves and Clinical Factors in Each Patient Group According to ARDI Change

Figure 3A shows that these five patient groups were significantly stratified for survival, with Groups D and E showing particularly poor prognosis (log-rank *p* = 0.04). When the entire patient population was divided into two larger groups of Group A + B + C and Group D + E, survival was significantly lower in Group D + E (log-rank *p* = 0.002, Figure 3B).

Table 2 compares patient backgrounds between Groups A–E, divided according to the pattern of ARDI change during the chemotherapy cycle by GBTM. Significant variations in age, initial treatment regimen, CCI, and GNRI were observed among the five groups. Table 3 shows a comparison of the patient backgrounds for Group A + B + C with good prognosis and Group D + E with poor prognosis. Thirty-six patients (11.7%) belonged to the two poor prognosis groups (Groups D and E). Patients in these poor prognosis groups were significantly older, more likely to have chosen R-THP-COP as the initial treatment regimen, had a lower tARDI throughout the entire treatment period, had a higher CCI, and had lower GNRI.

### 3.6. Clinical Factors Influencing ARDI Changes

Table 4 shows the results of multivariate logistic regression analysis for the selection of favourable prognosis groups as determined by GBTM (Group A + B + C). The results showed that patients with poor GNRI were significantly more likely to experience an unfavourable pattern of ARDI changes during the chemotherapy cycle (odds ratio 2.540, 95% confidence interval 1.020–6.310, *p* = 0.044).

## 4. Discussion

This study showed that a high ARDI in each cycle of standard regimens had a positive impact on the prognosis of de novo DLBCL up to the sixth cycle but may have a negative (rather than no) impact from the seventh cycle onwards. If the ARDI was less than 50% in the first half of the treatment period, the prognosis was poor, even though the ARDI increased in the second half of the standard regimens. Lower GNRI had a significant impact on patients falling into a changing pattern of ARDI during cycles of chemotherapy with poor prognosis.

The prognostic benefit of high ARDI for each cycle might be lost from the seventh cycle. This result is in line with previous findings that six cycles of R-CHOP/THP-COP-21 are not inferior to eight cycles [23,24,25,26]. The deceased group in our study was frailer and had a significantly lower total ARDI over the entire treatment period than the survival group, despite the significant aggressive lymphoma disease. According to the comparison of Cox-RCS models between the deceased and survival groups, low total ARDI in the deceased group might have resulted from the low ARDI in the initial cycle and a further reduction in ARDI after the second cycle. A higher total ARDI through the entire treatment period has a positive impact on the prognosis of DLBCL [3,4,5,6,7,8]. While tARDI is indisputably important for OS improvement, this principle appears to apply up to the sixth cycle, but not up to the eighth cycle.

According to the GBTM, two groups showed severely reduced ARDI with extremely low ARDI (e.g., below 50%) from the initial to the sixth cycle. Both of these groups displayed increased ARDI late through the entire treatment period, but still had significantly poorer prognosis than the other three groups. Once the ARDI falls very low, to below 50%, at any one time during the initial cycle or between the second and sixth cycles, raising the ARDI later in the treatment period may not lead to improved prognosis. We interpret this result as meaning that ARDI should be maintained as much as possible from the beginning of treatment until at least the sixth cycle. Besides the importance of the RDI of ADR during the first 12 weeks of therapy emphasised in a previous report [4], the ARDI of the first cycle of chemotherapy is also reportedly important in the prognosis of elderly patients with DLBCL [8]. Based on a previous report showing that patients without accumulation in interim 18F-fluoro-2-deoxy-D-glucose-positron emission tomography/computed tomography after 3 or 4 cycles have a more favourable prognosis than those with remaining accumulation [27], achievement of a rapid, deep response may contribute to the improvement of OS. We consider this to be one reason why the survival benefits of maintaining a high ARDI in the early chemotherapy period outweigh those in the late chemotherapy period.

We found that poor GNRI, a host-dependent factor rather than a disease-dependent factor such as LDH or stage, was significantly involved in the ARDI trajectory with poor prognosis. Our previous studies have identified host-dependent factors such as age and the presence of cognitive impairment as factors contributing to a reduction in the total ARDI to below 50% in elderly patients aged 80 years or older with DLBCL [9]. Further, we have recently reported that the host-dependent factor GNRI significantly influences the ability to receive standard treatments [16]. Considering the results of multivariate logistic regression modelling in the present study, factors contributing to prognosis by influencing the determination of treatment intensity are anticipated to be the host-dependent factor of nutritional status, rather than disease-dependent factors such as LDH or stage. Malnutrition is likely to have a negative impact not only on significant reductions in the initial dose of chemotherapy in malignancies, but also on significant delays or reductions in each cycle throughout the entire treatment period [28,29,30]. In groups such as cluster D, where treatment could be started with an ARDI of 80% for the initial cycle, but where ARDI would drop significantly after the second cycle, a postponement or dose reduction due to adverse events would be expected.

Although averaged/cumulative RDI is a well-known indicator for DLBCL prognosis [4,6], our findings suggest that individual patterns of ARDI change starting from the early treatment period provide additional information on the risk of death. In particular, reductions in ARDI within the first half of the treatment period were strongly associated with OS. The current study demonstrates that not only is cumulative/averaged RDI important, but certain populations, such as those with extremely low ARDI (e.g., below 50%) during the early treatment period, are more likely to experience diminished OS, reflecting higher risk. A better understanding of the differential trajectories of ARDI in DLBCL is important to determine subgroups at higher risk of mortality, the critical periods for maintaining ARDI, and minimal ARDI thresholds to optimise the effectiveness of immunochemotherapy. Given that fluctuations in ARDI are frequent, may compromise the benefits of chemotherapy for DLBCL, and can be addressed through evidence-based interventions [31,32], a better understanding of ARDI trajectories may help with the early identification of deteriorating patients and has potential implications for the personalised prevention of reduced ARDI.

Our study has several limitations that need to be kept in mind. First, this retrospective analysis was conducted at tertiary institutions, resulting in selection and healthcare access biases. The attending physician determines the overall treatment approach, considering not only disease-dependent factors such as disease status and IPI score, but also host-dependent factors such as age, PS, and frailty. Patients who are deemed able to tolerate high treatment intensity may have benefited from chemotherapy. Our study included a large number of elderly and vulnerable patients, so not all of these patients would derive equal benefit from a higher intensity of treatment. Second, the sample size was moderate, which has some implications for the analytical statistical power. We restricted eligibility to patients who had received at least six cycles of standard regimens. This restriction allowed us to exclude bias due to the inclusion of cases with different treatment strategies, such as limited-stage DLBCL. Third, results from the assessment of treatment responses after 4 cycles of treatment were not included in the database in the present study. One possibility is that treatment responses after 4 cycles of treatment may affect subsequent treatment intensity. This point necessitates further data collection and additional research. Finally, this research involved a region of Japan, currently the most super-aging society in the world. Our results thus may not be directly applicable to other countries and regions. However, given the aging population worldwide [33], universal implications appear likely in the near future.

Our study aims to reflect real-world clinical practice, where treatment intensity is determined at the discretion of the treating physician. Although our study included a certain number of elderly patients among the participants, many patients are targeted to maintain as high an ARDI as possible; even for patients over 80 years of age, there are a certain number of cases in which the goal is to maintain an ARDI above 50% (e.g., 80% to full dose). While phase II trials have demonstrated the safety and efficacy of R-miniCHOP, a 50% tARDI, the optimal tARDI for individual patients aged ≥80 years has not yet been determined. The therapeutic advantage of intensive chemotherapy close to full dose remains a subject of debate for patients aged ≥80 years, as noted elsewhere [9,34,35,36]. In a previous report, we presented real-world data on DLBCL patients aged ≥80 years in Japan, where 63% (80/127) of patients received standard regimens with total ARDI > 50% [9]. Although not all patients aged ≥80 years can tolerate higher total ARDI, certain populations may experience improved OS with higher treatment intensity. Properly selecting patient populations that will benefit from more aggressive treatment is crucial to prevent undertreatment. The present investigation aimed to reflect real-world clinical practice, where treatment intensity is determined at the discretion of the treating physician. We emphasise the need to individualise treatment decisions for elderly DLBCL patients, considering factors beyond age alone. Tailoring treatment intensity based on patient characteristics and overall health status is vital to optimise treatment outcomes and enhance the quality of life for these individuals.

## 5. Conclusions

In conclusion, maintaining a high ARDI in each cycle of standard regimens has a prognostic contribution to the outcome of de novo DLBCL up to the sixth cycle, but no effect from the seventh cycle. The negative prognostic impact of significantly lower ARDI in the early part of first-line chemotherapy could not be counteracted by increasing ARDI in the second half of the standard regimens. Patients with low GNRI presented with a significantly poor prognostic pattern of ARDI changes during cycles of chemotherapy.

## Figures and Tables

**Figure 1 cancers-15-04458-f001:**
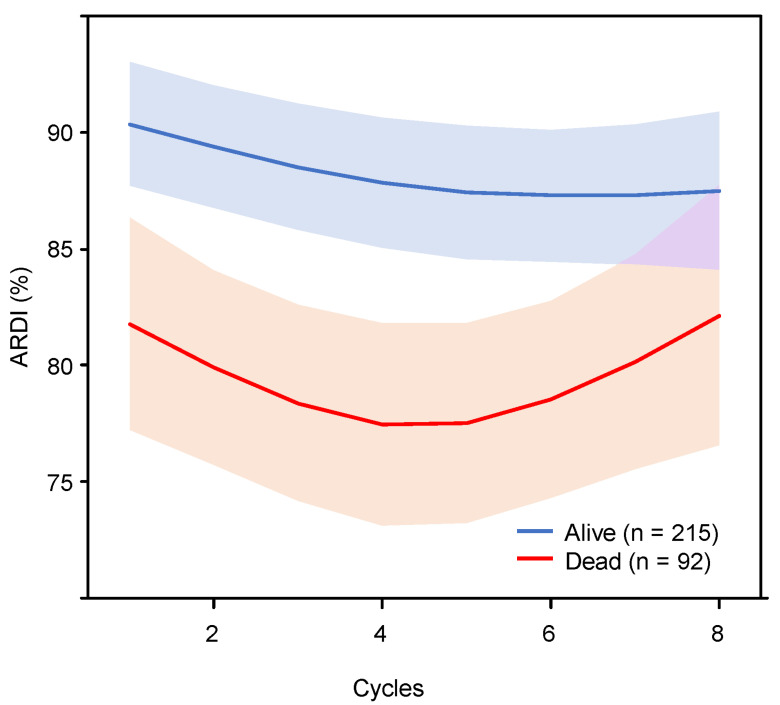
Association between average relative dose intensity during each chemotherapy cycle and all-cause mortality risk using a multivariate Cox hazards model with restricted cubic spline with 3 knots in each of the survival and deceased patient groups. The solid line represents the log hazards ratio, and the shaded area shows the 95% confidence interval. ARDI = average relative dose intensity.

**Figure 2 cancers-15-04458-f002:**
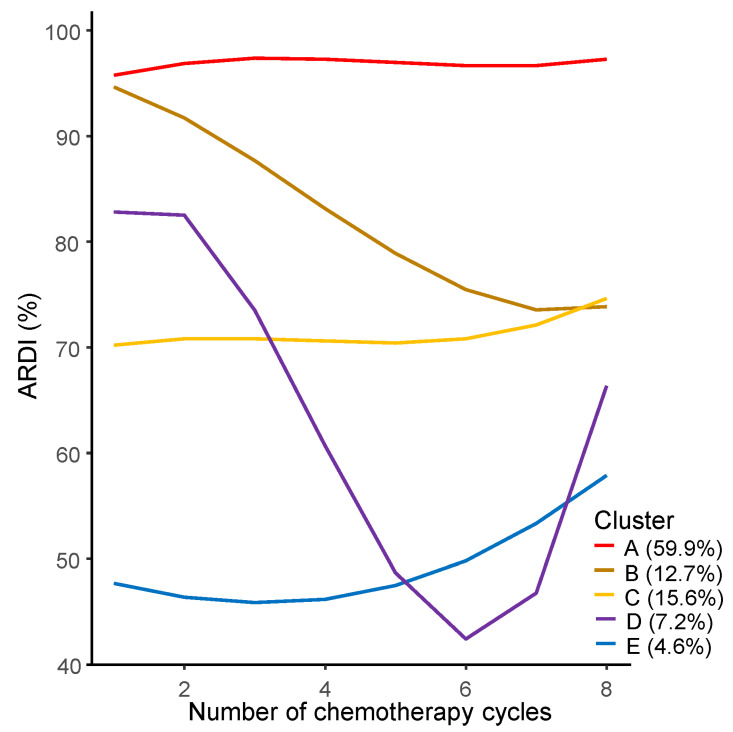
Group-based trajectory modelling to identify patterns of average relative dose intensity changes during the cycle of chemotherapy.

**Figure 3 cancers-15-04458-f003:**
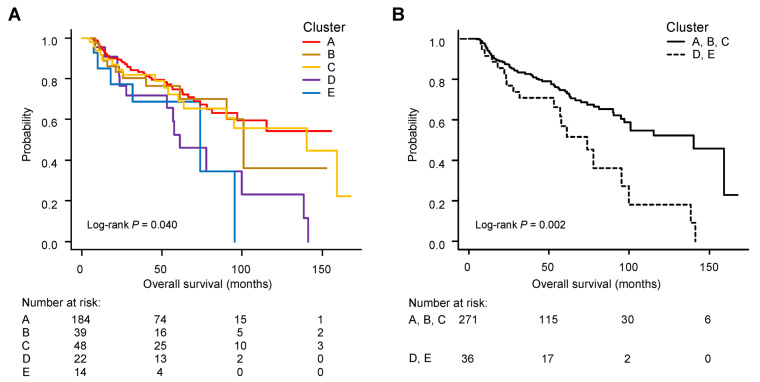
Kaplan–Meier curves for overall survival among patient groups according to group-based trajectory modelling. (**A**) Among five groups (Groups A–E). (**B**) Between two groups (Group A + B + C versus Group D + E).

**Table 1 cancers-15-04458-t001:** Patient characteristics by survival status at diagnosis.

	All Patients (*n* = 307)		Survivor Group (*n* = 215)		Deceased Group (*n* = 92)		*p* Value
Age, years—median, range	71	(16–96)	71	(16–89)	73	(46–96)	0.016
Age > 60 years	243	(79.2)	162	(75.3)	81	(88.0)	0.014
Male—*n* (%)	162	(52.8)	114	(53.0)	48	(52.2)	0.901
ECOG PS ≥ 2—*n* (%)	74	(24.1)	34	(15.8)	40	(43.5)	<0.001
LDH > ULN—*n* (%)	193	(62.9)	118	(54.9)	75	(81.5)	<0.001
Stage ≥ 3—*n* (%)	206	(67.1)	134	(62.3)	75	(81.5)	0.008
Extranodal sites ≥ 2—*n* (%)	120	(39.1)	75	(34.9)	45	(48.9)	0.022
IPI—*n* (%)							
Low	67	(21.8)	60	(27.9)	7	(7.6)	
Low-intermediate	63	(20.5)	51	(23.7)	12	(13.0)	<0.001
High-intermediate	74	(24.1)	51	(23.7)	23	(25.0)	
High	103	(33.6)	53	(24.7)	50	(54.4)	
B symptoms—*n* (%)	84	(27.4)	51	(23.7)	33	(35.9)	0.035
Bulky mass—*n* (%)	70	(22.8)	37	(17.2)	33	(35.9)	<0.001
sIL-2R, U/mL—median, range	1400	(125–38,400)	1160	(125–31,000)	2423	(332–38,400)	<0.001
tARDI, %—median, range	100.0	(19.2–143.3)	101.2	(35.4–143.3)	91.6	(19.2–137.8)	<0.001
THP-COP as initial therapy—*n* (%)	35	(11.4)	20	(9.3)	15	(16.3)	0.168
CCI score—median, range	1	(0–7)	1	(0–7)	1	(0–7)	0.012
GNRI score—median, range	95.9	(42.5–128.8)	98.7	(51.5–128.8)	90.3	(42.5–120.8)	<0.001

CCI, Charlson Comorbidity Index; ECOG PS, Eastern Cooperative Oncology Group performance status; GNRI, Geriatric Nutritional Risk Index; IPI, International Prognostic Index; LDH, lactate dehydrogenase; sIL-2R, soluble interleukin-2 receptor; tARDI, total all relative dose intensity; THP-COP, tetrahydropyranyl adriamycin, cyclophosphamide, vincristine, and prednisolone; ULN, upper limit of normal.

**Table 2 cancers-15-04458-t002:** Patient characteristics by clusters according to group-based trajectory modelling at diagnosis.

	All Patients (*n* = 307)		Cluster A (*n* = 184)		Cluster B (*n* = 39)		Cluster C (*n* = 48)		Cluster D (*n* = 22)		Cluster E (*n* = 14)		*p* Value
Age, years—median, range	71	(16–96)	67	(16–96)	74	(52–88)	77	(69–89)	83	(59–90)	74	(53–86)	<0.001
Age > 60 years	243	(79.2)	128	(69.6)	34	(87.2)	48	(100.0)	21	(95.5)	12	(85.7)	<0.001
Male—*n* (%)	162	(52.8)	103	(56.0)	14	(35.9)	26	(54.2)	12	(54.6)	7	(50.0)	0.252
ECOG PS ≥ 2—*n* (%)	74	(24.1)	39	(21.2)	12	(30.8)	10	(20.8)	10	(45.5)	3	(21.4)	0.121
LDH >ULN—*n* (%)	193	(62.9)	115	(62.5)	26	(66.7)	29	(60.4)	14	(63.6)	9	(64.3)	0.987
Stage ≥ 3—*n* (%)	206	(67.1)	124	(67.4)	25	(64.1)	30	(62.5)	16	(72.7)	11	(78.6)	0.805
Extranodal sites ≥ 2—*n* (%)	120	(39.1)	73	(39.7)	19	(48.7)	11	(22.9)	11	(50.0)	6	(42.9)	0.080
IPI—*n* (%)													
Low	67	(21.8)	46	(25.0)	6	(15.4)	11	(22.9)	3	(13.69	1	(7.1)	
Low-intermediate	63	(20.5)	38	(20.7)	9	(23.1)	10	(20.8)	2	(9.1)	4	(28.6)	0.569
High-intermediate	74	(24.1)	44	(23.9)	8	(20.5)	13	(27.1)	5	(22.7)	4	(28.6)	
High	103	(33.6)	56	(30.4)	16	(41.0)	14	(29.2)	12	(54.6)	5	(35.7)	
B symptoms—*n* (%)	84	(27.4)	52	(28.3)	7	(17.9)	16	(33.3)	5	(22.7)	4	(28.6)	0.573
Bulky mass—*n* (%)	70	(22.8)	44	(23.9)	12	(30.8)	6	(12.5)	5	(22.7)	3	(21.4)	0.300
sIL-2R, U/mL—median, range	1400	(125–38,400)	1429	(125–38,400)	1374	(184–14,871)	1134	(239–28,225)	1820	(291–21,263)	1570	(416–19,054)	0.532
tARDI, %—median, range	100.0	(19.2–143.3)	121.9	(85.0–143.3)	103.7	(77.1–120.1)	78.1	(59.1–105.7)	53.1	(19.2–74.1)	75.2	(48.8–96.6)	<0.001
THP-COP as initial therapy—*n* (%)	35	(11.4)	7	(3.8)	2	(5.1)	13	(27.1)	11	(50.0)	2	(14.3)	<0.001
CCI score—median, range	1	(0–7)	1	(0–6)	1	(0–4)	1	(0–7)	2	(0–7)	0	(0–5)	0.007
GNRI score—median, range	95.9	(42.5–128.8)	98.1	(51.5–125.3)	100.4	(80.3–128.8)	92.4	(60.1–113.4)	91.2	(42.5–120.0)	94.0	(79.5–106.0)	0.011

CCI, Charlson Comorbidity Index; ECOG PS, Eastern Cooperative Oncology Group performance status; GNRI, Geriatric Nutritional Risk Index; IPI, International Prognostic Index; LDH, lactate dehydrogenase; sIL-2R, soluble interleukin-2 receptor; tARDI, total all relative dose intensity; ULN, upper limit of normal.

**Table 3 cancers-15-04458-t003:** Comparison of patient characteristics when the five groups divided by group-based trajectory modelling are further divided into two groups according to prognosis.

	All Patients (*n* = 307)		Cluster A, B, C (*n* = 271)		Cluster D, E (*n* = 36)		*p* Value
Age, year—median, range	71	(16–96)	70	(16–96)	81	(53–90)	<0.001
Age > 60	243	(79.2)	210	(77.5)	33	(91.7)	0.051
Male—*n* (%)	162	(52.8)	143	(52.8)	19	(52.7)	0.999
ECOG PS ≥ 2—*n* (%)	74	(24.1)	61	(22.5)	13	(36.1)	0.095
LDH > ULN—*n* (%)	193	(62.9)	170	(62.7)	23	(63.9)	0.999
Stage ≥ 3—*n* (%)	206	(67.1)	179	(66.1)	27	(75.0)	0.347
Extranodal sites ≥ 2—*n* (%)	120	(39.1)	103	(38.0)	17	(47.2)	0.363
IPI—*n* (%)							
Low	67	(21.8)	63	(23.3)	4	(11.1)	
Low-intermediate	63	(20.5)	57	(21.0)	6	(16.7)	0.204
High-intermediate	74	(24.1)	65	(24.0)	9	(25.0)	
High	103	(33.6)	86	(31.7)	17	(47.2)	
B symptoms—*n* (%)	84	(27.4)	75	(27.7)	9	(25.0)	0.844
Bulky mass—*n* (%)	70	(22.8)	62	(22.9)	8	(22.2)	0.999
sIL-2R, U/mL—median, range	1400	(125–38,400)	1355	(125–38,400)	1815	(291–21,263)	0.093
tARDI, %—median, range	100.0	(19.2–143.3)	102.4	(59.1–143.3)	66.2	(19.2–96.6)	<0.001
Initial treatment with THP-COP therapy—*n* (%)	35	(11.4)	22	(8.1)	13	(36.1)	<0.001
CCI score—median, range	1	(0–7)	1	(0–7)	2	(0–7)	0.033
GNRI score—median, range	95.9	(42.5–128.8)	97.0	(51.5–128.8)	92.7	(42.5–120.0)	0.024

CCI, Charlson Comorbidity Index; ECOG PS, Eastern Cooperative Oncology Group performance status; GNRI, Geriatric Nutritional Risk Index; IPI, International Prognostic Index; LDH, lactate dehydrogenase; sIL-2R, soluble interleukin-2 receptor; tARDI, total all relative dose intensity; ULN, upper limit of normal.

**Table 4 cancers-15-04458-t004:** Multivariate logistic regression analysis for factors likely to present a pattern of favourable prognostic ARDI changes identified by group-based trajectory modelling.

	Odds Ratio (95%CI)		*p* Value
Age > 60 years	2.070	(0.576–7.400)	0.265
Male	1.040	(0.490–2.210)	0.920
ECOG PS ≥ 2	1.230	(0.551–2.760)	0.610
LDH > ULN	0.728	(0.319–1.660)	0.451
Stage ≥ 3	1.100	(0.431–2.810)	0.841
Extranodal sites ≥ 2	1.190	(0.539–2.620)	0.668
CCI category	1.420	(0.932–2.170)	0.102
GNRI risk	2.540	(1.020–6.310)	0.044

CCI, Charlson Comorbidity Index; CI, confidence interval; ECOG PS, Eastern Cooperative Oncology Group performance status; GNRI, Geriatric Nutritional Risk Index; LDH, lactate dehydrogenase; ULN, upper limit of normal.

## Data Availability

Data cannot be shared according to the protocol of the research participating institutions.

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
