# Peer review of "The Relative Dose Intensity Changes during Cycles of Standard Regimens in Patients with Diffuse Large B-Cell Lymphoma"

_cancers, 2023, doi:10.3390/cancers15184458_

Round 1

Reviewer 1 Report

Lee ate al reported a retrospective analysis on the changes of average relative dose intensity (ARDI) in a series of 307 DLBCL patients. They identified five groups based on the pattern of ARDI changes and those who had an ARDI fall to less than 50% has a poor outcome despite increased ARDI in the last part of therapy and those with a higher ARDI maintained throughout six courses of therapy, mainly early in the course of treatment, had a more favorable outcome.  The manuscript is interesting however there are several pitfalls that should be addressed as follows:

Patients enrolled were DLBCL treated with CHOP or THP-COP with or without Rituximab. The authors did not show any data regarding how many patients received Rituximab or not neither the impact of Rituximab on the outcome incorporating as a variable in multivariate analysis. The addition of Rituximab to chemotherapy has a strong prognostic vale on overall survival and it is now out of date to consider patients treated without Rituximab because no informative messages are given to the readers. I think that the analyses should be done only on patients treated with Rituximab taking out the others.

Diagnosis of DLBCL was made based on the old REAL classification and with WHO classification. Please state how many patients were classified with each one and also consider adding as covariable in the COX-RCS models because patients classified with the old REAL classification have a high risk to be misclassified. 

Patient characteristics. Please double check the text because it is not consistent with the table. For instance, patients in the survivor group did not have higher LDH, higher IPI, mor frequent bulky mass etc but just the opposite.

92 patients died but only 53 because of lymphoma. It means that 42.4% of all deaths were due to other causes. Please state the causes of death. This is an important point because the influence of ARDI changes on the outcome is crucial if they affect the lymphoma treatment and not if patients died for other reasons. Do you have data on Progression free survival? It would be nice to see if ARDI changes has a prognostic value also on PFS to reinforce the key messages.

In the multivariate Cox-RCS models IPI was not used as covariable. Why?

Same observation for the multivariate regression analysis showed in table 2

Figure S2A should be moved in the main manuscript or the data reported in the text

The text and many sentences are difficult to properly understand and a language revision by a native English speaker is advisable. 

The text and many sentences are difficult to properly understand and a language revision by a native English speaker is advisable. 

Reviewer 2 Report

In the manuscript titled “The relative dose intensity changes during cycles of standard regimens in patients with diffuse large B-cell lymphoma”, Lee et al. performed a study focused on the trajectory of average the ARDI during cycles of first-line chemotherapy in DLBCL. The highlight of this study is stratifying the fluctuations in ARDI as the chemotherapy cycles progress and focusing on the association with OS. The authors concluded that: 1) Maintaining a high ARDI in each cycle of standard regimens has a prognostic contribution to the outcome of de novo DLBCL up to the sixth cycle, but no effect from the seventh cycle; 2) The negative prognostic impact of significantly lower ARDI in the early part of first-line chemotherapy could not be counteracted by increasing ARDI in the second half of the standard regimens; 3) Patients with low GNRI presented with a significantly poor prognostic pattern of ARDI changes during cycles of chemotherapy. The group-based trajectory modelling (GBTM) is a vivid data interpretation way. However, I don’t think this study provided enough valuable information for our readers. I suggest this article should be rejected. Here are the comments as follows:

1. It seems that the authors were not clear with their data. Apparently, in results, part 3.1, the description of the results was not inconsistent with the data in the table. Patients in the survival group were with significantly lower LDH, lower number of extranodal lesions, lower IPI, less frequent presence of bulky mass, et al.

2. Likewise, data in supplementary table 2 could also be wrong. The numbers in columns “Cluster A, B, C” and “Cluster D, E” should be exchanged. These mistakes in the table and data interpretation implied that the writing of this manuscript was rough and far less precise.

3. In Cluster D, patients started the initial treatment cycle with an ARDI of approximately 80% and exhibited a steep decline in ARDI to less than 50% between cycles 2 and 6, but increased ARDI after cycle 6. Why did this group of patients increase ARDI after cycle 6? This seemed unreasonable. Did these patients achieve CR or PR after cycle 6? If patients did not achieve CR or PR after cycle 6, it is highly likely that increased intensity of regimens won’t work.

4. The assessment of treatment response after cycle 4 is important for the further treatment option and is predictive of clinical outcomes. However, no information about CR/PR state of patients enrolled was mentioned in the article.

Extensive editing of English language required

Reviewer 3 Report

The manuscript "The relative dose intensity changes during cycles of standard regimens in patients with diffuse large B-cell lymphoma" examined the ARDI and its impact on the outcome of patients with DLBCL

1. Would the authors suggest why the survival benefits of maintaining a high ARDI in the early chemotherapy period outweigh those in the late chemotherapy period in the Discussion?

2. "Dose modifications and the timing of the start of subsequent cycles were decided at the discretion of the attending physician" Could the authors comments how this would introduce selection bias and confound the analysis?

3. This cohort's median age at baseline was 71 years (range, 16–96 years). Many centres would plan to administer R miniCHOP rather than RCHOP for patients >80 yo due to potential toxicities. Were all the patients planned for full dose RCHOP and did the authors account for this when calculating the ARDI?

4. Did the authors evaluate for PFS?

5. "Patients in the survival group were significantly younger, with better PS, higher LDH, greater number of extranodal lesions, higher IPI, more frequent presence of bulky mass, higher sIL-2R, higher total ARDI, lower CCI, and higher GNRI" This statement seemed to suggest the survival group had higher risk disease with higher LDH, greater extranodal lesions, and IPI etc which is counterintuitive and is contrary to Table 1. Please clarify.

6."Median duration of follow-up was 40.7 months (range, 4.4–167.8 months), during 179 which time 92 patients died (29.9%), including 53 deaths (57.6%) due to lymphoma." A considerable number of patients did not die from lymphoma - what were the causes of death?

7. In patients with refractory disease and therefore had their treatment ceased earlier - how were these patients analysed? Did the refractory disease contribute to their lower ARDI? In those cases, the lower ARDI was not the cause of their poor outlook, but as result of their primary refractory disease. 

8. Could the authors please clarify/explain Table 2? The data seemed to indicate that none of the other traditionally considered high risk factors eg advanced stage disease, high LDH etc had impact on the ARDI?

There are some grammatical errors eg : Simple Summary: The important role of relative dose intensity (RDI) of first-line chemotherapy in 16 improving the prognosis of diffuse large B-cell lymphoma has been widely known, but no studies have focused on the trajectory of average the average RDI (ARDI) during cycles...

Some of the units listed did not appear to be commonly used: PSL (100 mg/body orally or intravenously on days 1–5)

Could the authors please look through and modify as appropriate

Reviewer 4 Report

I would suggest including figure S2A would be helpful to demonstrate effect of clusters (figure 3 on OS) They could be included together and reflected on in the text. 

In section 3.4, if the supplemental data is needed, please include in the main paper, otherwise suggest editing/ shortening this section as  readers  may not be able to access this information - so please reword with this in mind.

I am unclear how many patients received >6 cycles chemo - please clarify. Similarly, I do not think figure 2 adds much and would place this in the supplemental  information -text 194-200 largely covers this.

Round 2

Reviewer 2 Report

The manuscript was generally finely revised according to the reviewers’ comments and suggestions. The authors’ previous studies showed that: 1) the GNRI was not only a predictor of OS but also remarkably improved the prognosis prediction accuracy when incorporated with the CCI, having the ability to stratify the prognosis of elderly DLBCL patients; 2) maintaining the higher tARDI can achieve a better outcome even in this vulnerable population in real-world practice. Age, dementia, elevated LDH, CCI, and IPI affected the physicians’ decision-making to reduce tARDI. In the present study, the authors tried to demonstrate that maintenance of ARDI particularly up to cycle 6 is important for prognosis, and GNRI is a determinant factor that influences ARDI changes. However, I don’t suppose that the conclusion from the present study provides profound insights beyond their previous studies. The group-based trajectory modelling could well identify subgroups of individuals with similar trajectories. However, it was not a novel finding that patients in clusters D and E who received lower ARDI displayed poorer prognosis. Patients in clusters D and E present about 10% of total patients, and the comparison made between these patients and the rest majority might cause a bias. Besides, this means this minority of patients received this treatment approach for some specific reasons likely not only due to lower GNRI. Likewise, in aged patients, GNRI should be taken into account when the attending physician determines the treatment plan, leading to different trajectories. This point was also raised in the authors’ previous study. All in all, I think this manuscript didn’t provide enough information to our readers and I suggest this article should be rejected.

no comment

Reviewer 3 Report

Thanks for the authors' responses and modifications. They have addressed the concerns. 

 Could the authors please review and edit English language as required